# EAGLE: Egocentric AGgregated Language-video Engine

## ABSTRACT

The rapid evolution of egocentric video analysis brings new insights into understanding human activities and intentions from a first-person perspective. Despite this progress, the fragmentation in tasks like action recognition, procedure learning, and moment retrieval, etc., coupled with inconsistent annotations and isolated model development, hinders a holistic interpretation of video content. In response, we introduce the EAGLE (Egocentric AGgregated Language-video Engine) model and the EAGLE-400K dataset to provide a unified framework that integrates various egocentric video understanding tasks. EAGLE-400K, the *first* large-scale instruction-tuning dataset tailored for egocentric video, features 400K diverse samples to enhance a broad spectrum task from activity recognition to procedure knowledge learning. Moreover, EAGLE, a strong video-based multimodal large language model (MLLM), is designed to effectively capture both spatial and temporal information. In addition, we propose a set of evaluation metrics designed to facilitate a thorough assessment of MLLM for egocentric video understanding. Our extensive experiments demonstrate EAGLE's superior performance over existing models, highlighting its ability to balance task-specific understanding with comprehensive video interpretation. With EAGLE, we aim to pave the way for novel research opportunities and practical applications in real-world scenarios.

## CCS CONCEPTS

• **Computing methodologies → Planning with abstraction and generalization**; **Planning for deterministic actions**; *Planning under uncertainty*; **Neural networks**; Search with partial observations; **Image and video acquisition**; **Natural language generation**; **Information extraction**; **Discourse, dialogue and pragmatics**; **Temporal reasoning**; **Spatial and physical reasoning**; **Computer vision problems**; **Computer vision representations**; **Computer vision**.

## KEYWORDS

Augmented Reality, Egocentric Video Analysis, Integrated Video Understanding Framework, Egocentric Video Dataset, Spatial and Temporal Information Processing, Multimodal Large Language Models (MLLMs), Comprehensive Video Interpretation, Performance Evaluation Metrics

## 1 INTRODUCTION

Understanding human activities and intentions in videos is a key challenge for intelligent systems, requiring advanced reasoning capacities. While there have been advancements in computer vi-

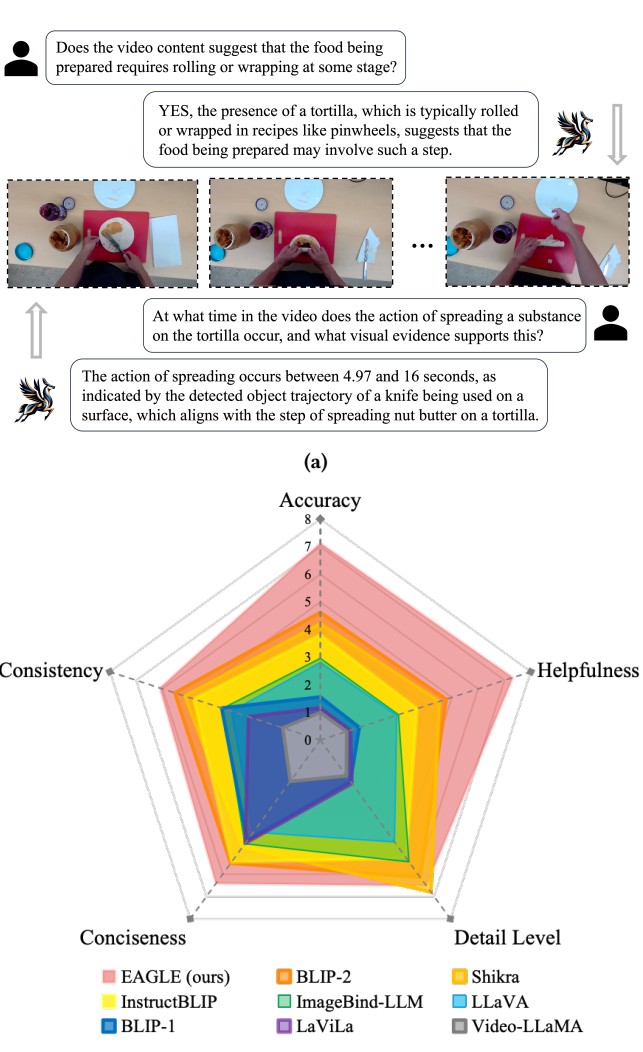

Figure 1: (a) illustrates the EAGLE, a framework designed to unify egocentric video tasks, thereby facilitating inter/intra-task understanding. (b) shows evaluation results of existing methods, including our EAGLE model and BLIP-2 [43], BLIP-1 [44], InstructBLIP [15] etc., using the newly proposed metrics on the EAGLE-400K benchmark.

sion, the most notable breakthroughs are seen in the evolution of Large Language Models (LLMs) [14, 63]. These models benefit from increased data and model size, resulting in enhanced

generalizability, which is often challenging to achieve in computer vision tasks. By leveraging the pre-trained LLMs [21, 106], MLLMs [9, 12, 15, 25, 29, 43, 44, 50, 101, 105] show superior results to a wide spectrum of multimodal tasks [22, 34, 39, 58, 59, 77, 81]. Unlike current MLLMs that predominantly focus on images, EAGLE advances to capture spatial and temporal information to enable more in-depth video analysis. To enable MLLM to achieve a more holistic and detailed examination of human activities, our work pivots from previous efforts focused on third-person view [10, 19, 37, 40, 82], towards the egocentric view, offering an unfiltered and untrimmed perspective. This perspective enhances exocentric tasks like action recognition and localization by offering insights into individual interactions with their surroundings and facilitates new unique tasks like Natural Language Queries and Action Anticipation [23]. These tasks demand an in-depth view of the video content, including activity recognition and procedure knowledge learning. Taking sandwich preparation as an example, the task requires recognizing actions like preparing ingredients and spreading condiments and understanding how these actions contribute to the overall process. Pioneering efforts like EPIC-KITCHENS-100 (EPIC-KITCHENS) [38] and Ego4D [23] have paved the way for tasks focused on activities like temporally localizing and anticipating actions. Subsequent research [5, 8, 74] has extended these concepts by introducing tasks that emphasize procedure knowledge, aiming to understand actions' intentions and contextual relevance.

While diverse tasks offer more insights, they also foster task-specific models, similar to traditional approach in the NLP field, where models are trained for specific tasks like sentiment analysis, translation, and question-answering, etc. This results in a fragmented approach, where each model specializes in a specific aspect. For instance, one model may excel in recognizing actions timestamp (e.g., identifying a *'gra b a spoon'* from seconds 5-7), while another pinpoints the timing of such actions. These two example tasks, though different in focus—action recognition versus temporal localization—essentially seek to identify the action and its temporal occurrence. Many works [33, 35, 36, 54] have attempted to mitigate these problems by employing a shared backbone [68, 75] or re-scaling labels [48, 99]. These approaches are limited by their reliance on task-specific models, highlighting the challenge in egocentric video understanding: balancing specialization with a holistic grasp of video content.

Addressing the above challenges, we introduce the EAGLE-400K dataset, the *first* large-scale instruction-tuning dataset tailored for egocentric video. We provide a unified task interface that not only integrates existing tasks but also fosters the development of new, context-rich tasks as shown in Table 3 Compared with existing large image-based instruction tuning (LLaVA-150K [50], VideoInstruct100K introduced by Video-ChatGPT [55]), our method is *3-4×* times larger to facilitate the research field. This dataset is a comprehensive collection designed to advance the understanding of activities and procedure knowledge in an egocentric view. It comprises 36k video clips sourced from three different origins: Ego4D and EPIC-KITCHENS, which are for activity recognition, and PTA, which is crucial for procedure learning, as detailed in Table 4. By employing instruction tuning, EAGLE-400K unified fragmented tasks as coherent (VIDEO, INSTRUCTION, RESPONSE) pairs, thereby serving as a high-quality, large-scale video instruction

tuning dataset, as shown in Table 3. Moreover, EAGLE-400K leverages existing annotations to facilitate knowledge sharing across datasets, which enables the creation of novel tasks, such as Temporal Reasoning and Cross-Referencing Events as shown in Table 3that were not present in the original dataset.

Complementing the dataset, we propose EAGLE, a video-based MLLM, we augment its capacity for spatial and temporal reasoning through the integration of the Adapter [28] We conducted a systematic evaluation to demonstrate the efficacy and adaptability of the proposed dataset and model, comparing EAGLE with leading MLLMs, including BLIP-2 [43], BLIP-1 [44], InstructBLIP [15], LaViLa [105], LLaVA [50], ImageBind-LLM [25], Shikra [12], Video-LLaMA [102]. The results, as illustrated in Figure 1b, EAGLE outperforms all models on the proposed benchmark. We summarize our main contributions as follows:

- **EAGLE-400K Dataset:** Our work introduces the pioneering large-scale video instruction-tuning dataset for egocentric video understanding [83], providing a unified task interface to alleviate the models and task fragmentation. At *4×* times the size of the previously largest video instruction-tuning dataset, EAGLE-400K is expected to greatly benefit the community by encouraging further novel research and serving as a benchmark for evaluation.
- **PTA dataset** To fill the gap in procedural understanding within current egocentric video datasets, we have collected and annotated the Perception-driven Task Assistance (PTA) dataset.This dataset contains 268 egocentric videos, each recorded with certain recipe scripts to provide a rich, detailed insight into specific procedural tasks, advancing our understanding of egocentric procedures.
- **EAGLE Model.** We introduced the EAGLE model, a novel video-based MLLMs designed to excel at capturing both spatial and temporal information with the advantage of EAGLE-400K. To the best of our knowledge, we are the first to incorporate fine-grained object trajectories, temporal boundaries, and scripted procedure videos for *video instruction tuning*.
- **Evaluation Metrics.** we provide an comprehensive analysis of current state-of-the-art MLLMs to highlight their limitations and the challenges of applying them to egocentric video understanding.e proposed a novel metric designed to offer a more comprehensive assessment, further demonstrating our model's superior performance on the proposed benchmark.

## 2 RELATED WORK

### 2.1 Egocentric Video Understanding

Egocentric Video Understanding began with pioneering datasets [17, 46, 66] that demonstrated the unique potential of first-person video analysis. The field expanded with EPIC-KITCHENS [16], featuring 100 hours of videos, and further with Ego4D [24], which boasts an impressive 3,000 hours of data. These expansions inspired a wide range of research tasks, including human-object interactions [60, 97], activity recognition [38, 69, 87, 93], sounding object localization [1, 31, 32, 56, 96, 107], pose estimation and prediction [6, 62, 89], procedure knowledge learning [5, 26], and social understanding [76]. However, various tasks have resulted in specialized, fragmented model development. EAGLE-400K aims to consolidate these tasks for a more holistic video understanding.

## 2.2 LLMs for Multimodal Understanding

Recent advancements have extended LLMs to multimodal domains, resulting in MLLMs [2, 15, 20, 42, 44, 47, 102, 109] that excel in various tasks. Fine-grained multimodal Understanding involves a detailed understanding of visual content, including spatial details [11, 12, 41, 65, 90, 94, 98, 101, 104], temporal sequences [48, 48, 64, 84, 85, 91, 92], or a combination of both [7, 51, 88]. Models like [15, 44, 109] use a two-stage Q-former to align vision and language models. [102] aligns video and audio modalities with LLMs by training adapters, showing its ability to integrate multiple modalities effectively. Video-ChatGPT [55] and VideoChat [45], combining LLMs with video foundation models, are tailored for coarse-grained video-based conversations. However, few MLLMs are designed to tackle both spatial and temporal video tasks [83], and our work emphasizes interpreting 16 seconds videos, which are *2-4×* longer compared with other video MLLMs.

## 2.3 Fine-grained Multimodal Comprehension

Fine-grained multimodal comprehension involves a detailed understanding of image or video content, including spatial [11, 12, 41, 65, 90, 94, 98, 104], temporal [48, 48, 64, 79, 84, 91, 92], or both spatial and temporal [7, 88] information. The multimodal models for fine-grained spatial understanding like [12] and [98] are utilizing LLMs trained on an instruct-tuning dataset which is produced by the language-only GPT-4 and include the coordinates of objects' bounding boxes. They can handle multiple location-related multimodal tasks like REC, PointQA, dense image captioning, and VQA. In [65, 104], special tokens representing the regions are used, while [11] and [3] adopt both special tokens and coordinates. [41, 90] implemented irregular pixel-level region segmentation, generating descriptive captions for any object within an image. The multimodal models for fine-grained temporal video understanding, including [48, 64, 85], are leveraging the capabilities of LLMs. There are seldom multimodal models designed to handle both spatial and temporal video understanding tasks.

## 3 EAGLE-400K DATASET AND BENCHMARK

Egocentric video understanding [61, 67] involves two primary aspects: *activity recognition*, which identifies individual actions like picking up objects, and *procedure knowledge learning*, which models sequential action relations to understand their contribution to achieving a goal. We aim to *consolidate multiple datasets* with different focuses and provide a comprehensive dataset. We start with two popular egocentric datasets, EPIC-KITCHENS [38] and Ego4D [23], featuring long-term, untrimmed videos of daily tasks. These datasets are annotated with natural human actions and object interactions without predefined procedures, focusing on solely on identifying actions.

However, these existing datasets provide only action labels without encapsulating procedure knowledge. To bridge this gap, we've also gathered the PTA dataset, consisting of 268 egocentric videos recorded in laboratory settings. This dataset is specifically designed to enhance procedure knowledge learning through detailed visualization of three distinct recipes: pinwheel, mug cake, and broil coffee. Unlike previous approaches [5, 78] which prioritized task diversity but lacked depth within individual tasks, our approach focuses

Table 1: The table compares vision-language instruction-tuning datasets, including EAGLE-400K and MIMIC-IT. MIMIC-IT generates questions from visual descriptions but often produces questions not closely related to the visual content due to noisy narration. VideoInstruct is generated from ActivityNet-200 [27] and serves as popular video instruction tuning dataset, featuring short clips paired with QA-style data without spatial-temporal understanding.

| Dataset | Video | #Clip | #Ins. | #Ins./clip | Duration |
|---|---|---|---|---|---|
| MiniGPT-4 [109] | × | - | 5K | - | - |
| Shikra-RD [12] | × | - | 5.9K | - | - |
| LLaVA [50] | × | - | 345K | - | - |
| MIMIC-IT [42] | ✓/× | 400K | 2.4M | 6 | 4-8 frames |
| VideoInstruct [45] | ✓ | 13k | 100k | 7 | 5 s |
| **EAGLE-400K** | ✓ | 36K | 400K | 11 | 16-76s |

on providing extensive variation and a higher number of samples within a select few tasks. This approach enables a more comprehensive analysis of procedural steps, making the PTA dataset a valuable resource. Representative examples from PTA dataset are shown in Figure 2.

We split the data into training and validation sets according to established splits for Ego4D and EPIC-KITCHENS. For PTA, we used a 70/30 split, excluding videos from one lab to serve as a novel testing environment. The remaining testing videos were randomly sampled as detailed in Table 4.

## 3.1 Annotation

For EPIC-KITCHENS split, we utilized official annotations that include action-object labels with temporal boundaries as shown in Figure 2. Additionally, we integrated spatial annotations from the EPIC-KITCHENS-VISOR dataset [18], an extension of EPIC-KITCHENS, providing object segmentation trajectories covering one-third of the original EPIC-KITCHENS dataset. In the case of Ego4D, the initial ~3.8 million narrations underwent refinement to generate various subsets, as outlined in [24]. Our focus lies on the Episodic Memory and Forecasting Benchmark, which includes tasks such as Natural Language Queries, Moment Queries, and Long-term Action Prediction tasks, all tailored for activity understanding. In the PTA subset, each video depicts the process of making a recipe, with timestamps marked for key procedure steps.

To enrich the annotation with object information, we first fine-tuned the DINO [52] using the EgoObject dataset [108] without its class head, significantly improving its object proposal accuracy to over **90%** on the test set. Next, we integrated this enhanced DINO model with the latest DEVA [13] tracker, achieving reliable object tracking from an egocentric viewpoint. Lastly, we employed the LLaVA-13B model to interpret the semantic meanings of the proposed object regions. As shown in Figure 2, while this approach may not always reach the accuracy level of human annotation—occasionally mistaking a tortilla for flatbread, for instance—it marks a considerable leap forward, especially given the scarcity of zero-shot vision models capable of achieving high accuracy in grounding.

**Figure 2: Left: Representative frames from the Ego4D [23], EPIC-KITCHENS [16], and PTA datasets, showcasing the intricate capture of task-oriented activities. Right: Visualizations of trajectories and object interactions within the EAGLE-400K dataset, emphasizing the tasks' complexity and diversity.**

**Table 2: An example task on EAGLE-400K: the preparation of a dish called "Pinwheel" from PTG data. It details the dish preparation process and featured objects. The task involves placing various ingredients on a tortilla with a knife, performed by a participant wearing a camera. The top block presents prompts for GPT, including captions and object boxes, while the bottom block shows question types and responses. Notably, the visual image does not prompt GPT.**

---

**Context type 1: Task Description**

Pinwheels with steps 1: Place the tortilla on the cutting board., 2: Scoop nut butter and spread it on the tortilla, leaving a margin at the edge., 3: Clean the knife with a paper towel., 4: Using the knife, scoop jelly from the jar and spread it over the nut butter., 5: Clean the knife with a paper towel., 6: Roll tortilla into a tight, 1.5-inch thick log without squeezing out the filling, 7: Secure the roll with 5 toothpicks spaced 1 inch apart., 8: Trim tortilla roll ends, leaving a 1/2 inch margin near the last toothpick; discard the ends., 9: Place floss under the roll, halfway between two toothpicks, perpendicular to its length, 10: Cross floss ends over the roll and pulls in opposite directions to slice., 11: Continue slicing with floss to create 5 pinwheels., 12: Place the pinwheels on a plate..

The current step, as ground truth, is: <0,16> 4: scoop jelly and spread jelly

**Context type 2: Object Trajectory**

A jar of ice cream is sitting on a table.: [12, 0.215, 0.57],[4, 0.17, 0.57],[10, 0.2, 0.56],[7, 0.185, 0.545],[6, 0.175, 0.47],[8, 0.175, 0.52],[13, 0.16, 0.53],[14, 0.185, 0.695] A person is using a knife to spread peanut butter on a plate.: [12, 0.78, 0.61],[9, 0.755, 0.59],[11, 0.75, 0.59],[8, 0.735, 0.555],[6, 0.745, 0.52],[14, 0.73, 0.55],[7, 0.75, 0.57],[10, 0.755, 0.6] A bowl of peanut butter is being held by a person.: [11, 0.765, 0.68],[0, 0.71, 0.795],[4, 0.75, 0.65],[3, 0.675, 0.645],[13, 0.755, 0.675],[8, 0.735, 0.645],[12, 0.785, 0.705],[14, 0.755, 0.66] A man is using a knife to cut a piece of cheese on a plate.: [5, 0.79, 0.825],[8, 0.855, 0.81],[6, 0.85, 0.75],[12, 0.875, 0.905],[15, 0.84, 0.94],[14, 0.83, 0.88],[11, 0.905, 0.85],[10, 0.905, 0.895] A person is holding a bowl of yellow paint.: [6, 0.76, 0.6],[1, 0.87, 0.835],[4, 0.735, 0.63],[10, 0.76, 0.67],[5, 0.725, 0.615],[7, 0.77, 0.645],[11, 0.765, 0.675],[13, 0.74, 0.645]

---

**Response type 1: Task Verification**

**Question:** Based on the video content, is the person in the video currently spreading something on the tortilla, and if so, what is it?

**Answer:** YES, the person is spreading jelly on the tortilla. This is indicated by the action of using a knife to scoop out a substance, likely jelly, from a container and then spreading it onto a flat surface, which is consistent with step 4 of the recipe.

**Response type 2: Step Verification**

**Question:** At what moment in the video can you confirm that jelly is being spread on the tortilla, and what visual clues support this?

**Answer:** The action of spreading jelly on the tortilla can be confirmed throughout the video, as indicated by the trajectory of the knife moving across the frame, consistent with a spreading motion. The presence of a container from which the substance is being scooped also supports this.

**Question:** How does the movement of the knife in the video align with the current step of spreading jelly?

**Answer:** The movement of the knife in the video aligns with the spreading action required in the current step. The trajectory data shows the knife moving across the frame in a manner consistent with spreading a substance on a flat surface.

**Response type 3: Objects Verification**

**Question:** At what time and location in the frame does the jar of jelly appear, and how is it relevant to the task?

**Answer:** The jar of jelly appears throughout the video, with its location varying slightly but generally centered around the coordinates (0.2, 0.57). It is relevant to the task as it is the source of the jelly being spread on the tortilla, which is a key step in making pinwheels.

---

## 3.2 Instruction Tuning Data Generation

As previously mentioned, diverse tasks and inconsistent annotation standard often limit the comprehensive understanding of videos. We adapt the instruction tuning [100] to unify these annotations under a cohesive framework. In our dataset, videos are segmented into 16-second clips, *3-5×* longer than common video understanding dataset, ensuring each contains a rich number of actions while maintaining a manageable length, as shown in Table 4. By comparison, our baseline model, LaViLa [? ], which is trained specifically on egocentric videos, typically takes a 1-sec clip. Another example is EPIC-KITCHEN Action Anticipation task, although videos tend to

be minutes, only a 5-second segment is used for analysis. Adopting 16-second clips allows us to capture comprehensive action details without overwhelming the model.

To determine the optimal frame rate, we draw inspiration from recent studies [73, 95] that have shown promising results in frame-based video understanding by analyzing videos frame-by-frame and using feature pooling. Building on this, we sample one frame per second, maintaining a consistent interval regardless of the video's frame rate. To enhance contextual understanding, we incorporate temporal context with 30 seconds before and after each clip. We chose a 30-second duration to balance action details and cohesive

**Table 3: The table outlines activities in a kitchen video, including opening/closing cabinets, grabbing a knife, and washing vegetables, showcasing a person's kitchen work. It serves as an example of instruction-following data. The top block displays prompts like captions and boxes for GPT, while the bottom block shows response types. Notably, the visual image does not prompt GPT and is included for reference only.**

---

**Context type 1: Temporal History**

Past 30 second: take container, take tofu, close fridge, open fridge, take carrots and, open drawer, close fridge, put down vegetables, open cupboard, take cutting board, put down cutting board Current <0,0.76> close cupboard, <3.66,5.0> open drawer,<5.5,8.0> take knife, <5.55,6.36> take knife, <6.73,7.74> put down knife, <8.1,8.6> close drawer, <9.45,10.7> put down knife,<11.05,14.3> open tap,<13.86,16> wash courgette

Future 30 second: still washing courgette, wash carrot, still washing carrot, close tap, put down vegetables, open cupboard, take grater, take pan, put down pan, close cupboard, close cupboard, take courgette, cut courgette

**Context type 2: Object Trajectory**

'right hand': [[5.0, 0.295, 0.401], [6.0, 0.317, 0.419], [7.0, 0.294, 0.365], [8.0, 0.324, 0.406], [10.0, 0.303, 0.377], [12.0, 0.344, 0.366], [13.0, 0.336, 0.284]] 'left hand': [[5.0, 0.186, 0.407], [6.0, 0.207, 0.427], [7.0, 0.212, 0.402], [8.0, 0.214, 0.402], [11.0, 0.293, 0.368], [12.0, 0.113, 0.324], [13.0, 0.216, 0.358]] 'drawer': [[5.0, 0.258, 0.311], [6.0, 0.193, 0.319], [7.0, 0.227, 0.284], [8.0, 0.219, 0.31]] 'chopping board': [[9.0, 0.266, 0.108], [10.0, 0.203, 0.348], [11.0, 0.153, 0.373], [12.0, 0.045, 0.395], [13.0, 0.035, 0.404]] 'carrot': [[10.0, 0.218, 0.177], [11.0, 0.241, 0.233], [12.0, 0.096, 0.252], [13.0, 0.075, 0.264]]

---

**Response type 1: Description**

**Question:** Can you provide a concise summary of the main events occurring in the video?

**Answer:** Certainly. The video shows a sequence of kitchen activities, including opening and closing a cupboard and drawer, taking and putting down a knife, and washing a courgette.

**Response type 2: Detailed Description**

**Question:** Can you provide a detailed account of the sequence of events that take place between 3.66 and 8.6 seconds in the video, specifically focusing on the actions involving the kitchen drawer and knife? Please include the timing of these actions and describe how the person's hands interact with these objects, as indicated by the object trajectory data.

**Answer:** Between 3.66 and 8.6 seconds in the video, the individual engages in a sequence of actions involving a kitchen drawer and a knife. Initially, between [3.66, 5.0] seconds, the right-hand reaches for the drawer, with its trajectory indicating a movement slightly right and above the center, coinciding with opening the drawer. Then, the person takes a knife between [5.5, 6.36] seconds, as evidenced by the right hand moving downward and leftward towards the drawer. This action involves a brief re-adjustment, as the person takes the knife, adjusts their grip, and takes it again. There's a moment between [6.73, 7.74] seconds where the knife is put down, suggested by the right hand moving away from the drawer. Finally, the drawer is closed between [8.1, 8.6] seconds, with the right hand's trajectory returning to its original position, indicating the closing motion. Throughout this sequence, the left hand appears to provide support or stabilization, with minor adjustments in its position.

**Response type 4: Event Localization**

**Question:** At what timestamps does the individual open and close the drawer?

**Answer:** The individual opens the drawer between [3.66, 5.0] seconds and closes it between [8.1, 8.6] seconds in the video.

**Response type 5: Temporal Reasoning**

**Question:** How did the previous actions of taking vegetables and opening the fridge lead to the current situation in the video?

**Answer:** Previous actions like taking vegetables and opening the fridge led to the current situation by indicating food preparation activities, culminating in washing the courgette.

**Response type 6: Event Localization**

**Question:** Given the ongoing event of washing vegetables, what might be a likely future action sequence in the kitchen, and why?

**Answer:** Based on the ongoing event of washing vegetables, future actions may include chopping the courgette, possibly using the grater and pan (as indicated in the 'future' events list), suggesting a continuation of food preparation.

**Response type 7: Cross-Referencing Events**

**Question:** Can you explain the connection between opening the drawer (3.66-5.0 seconds) and the subsequent use of the chopping board (after 9.0 seconds)?

**Answer:** The opening of the drawer (3.66-5.0 seconds) and the use of the chopping board (after 9.0 seconds) are connected as both actions are part of setting up for the food preparation process; utensils are gathered first (from the drawer), followed by setting up the chopping board for cutting vegetables.

---

narration. This is based on our observation that longer durations reduce the relevance of actions. In this way, the context helps tasks like action anticipation and detection and encourages the development of new tasks by extrapolating relationships between labels. For instance, our framework enables advanced tasks such as Temporal Reasoning and Cross-Referencing Events, as shown in Table 3, enhancing the dataset's utility without additional annotation effort.

We use two types of symbolic representations to prompt GPT4: (i) Captions, which typically describe the visual scene from various perspectives. (ii) Objects trajectory in the scene, and each box encodes the object concept and its spatial location as shown in Figure 2. We collect 400K unique video instruction-following samples in total, including 350K for activity recognition as shown in Table 3 and 50K for procedure knowledge learning. We have undertaken multiple iterations to refine our method for creating accurate instruction data from task descriptions and object trajectories. We normalized object bounding boxes to a scale of 0-1 and used only the center points of objects, improving the spatial relationships in GPT-4's responses. Additionally, we added a post-processing step that uses

interpolation to align GPT-4 output coordinates with actual object trajectories, ensuring high data accuracy. However, including complete trajectories in responses sometimes led to errors. To counter this, we selectively replaced faulty segments with ground truth data, enhancing the dataset's usability. As shown in Table 1, our approach provides longer question-to-clip correspondence than MIMIC-IT [42], focusing on video content comprehension. In contrast, MIMIC [23] often generates questions unrelated to the visual content. Compared to EgoSchema [57], our method emphasizes fine-grained understanding, while EgoSchema targets coarse-grained analysis with few multiple-choice questions for 3-minute video.

## 4 EAGLE MODEL

Existing image-based MLLMs such as Shikra [12] primarily focus on spatial information, while models like VTimeLLM [30] specifically target temporal dimensions. Given the unique aspects of our dataset, which encompasses both spatial and temporal attributes, our goal is to simplify the tuning process and construct a straightforward yet strong model by leveraging the existing MLLM model.

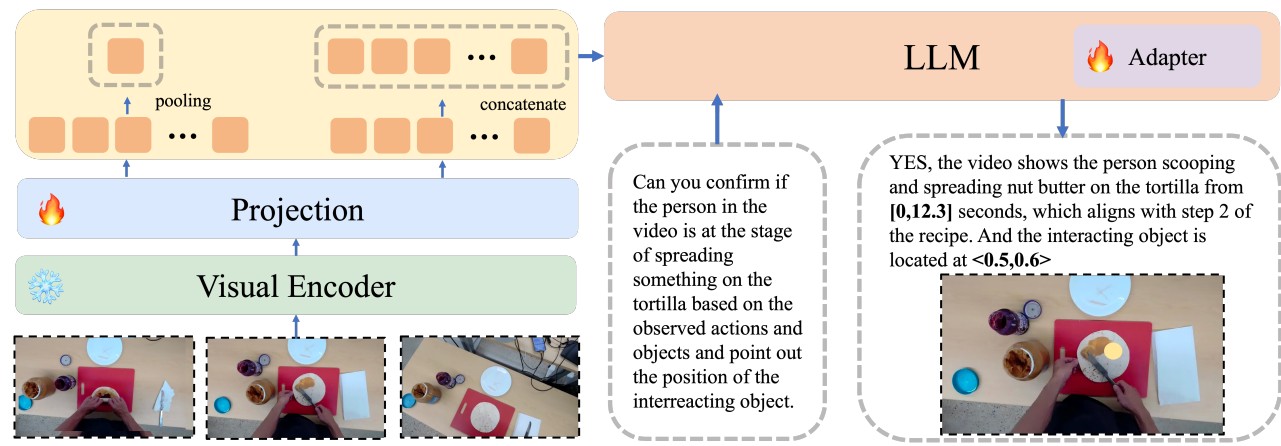

**Figure 3: The architecture of the EAGLE model, highlighting the fine-tuned projection layer and adapter, enhances the LLM's ability to process time boundaries and object location tokens. EAGLE is capable of processing natural language queries to determine the temporal boundaries of events and pinpoint the location of the desired object, as denoted by the yellow dot.**

Our model, in line with common MLLMs, integrates a vision encoder, an alignment layer, and a large language model (LLM), specifically employing the pre-trained ViT-L/14 from CLIP [70] as the frame encoder $\mathbb{E}$ and Vicuna-13B as LLM, as shown in Figure 3. Given a video sample $V_i \in \mathbb{R}^{T \times H \times W \times C}$ with $T$ frames, the frame encoder $\mathbb{E}$ processes each frame independently, generating video embedding as $x_i \in \mathbb{R}^{T \times D}$.

After obtaining frame embeddings, selecting an optimal method for aggregating these features is critical. Video-LLaMA [103] employs temporal position embedding and a q-former, which typically demands a large amount of paired video-text data (rare in video datasets). Compared with image-language datasets such as CC3M [80] utilized by LLaVA [49], video-language datasets like WebVid [4] contain shorter and less detailed language descriptions. Consequently, when models are pretrained on these video datasets, their expressiveness is often limited, which can result in a less effective image-language alignment layer. Instead, we choose to leverage existing alignment layers from LLAVA to obtain language token from visual feature. We have two streteges, (i) adapt recent advancements [73] and employ an average-pooling strategy to aggregate a video-level representation $v_i \in \mathbb{R}^D$, where D is 5,120 for Vicuna-13B. We donate this model as EAGLE-pool. (ii) Instead of using pooling, we employ alignment layers to extract language tokens directly from each visual frame and concatenate these tokens into a long sequence. This method does not require the explicit embedding of position tokens. Instead, it implicitly incorporates temporal learning, thus leveraging the strengths of the LLAVA alignment layer, which ensures more reliable alignment compared to Q-former aggregation methods.

To enhance the LLM's ability to capture both temporal and spatial information, we integrated Adapters [28] into various self-attention layers of Vicuna-13B, allowing the model to effectively incorporate coordinates from both time boundaries and object trajectories. During training, the visual embedding can be inserted anywhere in the input sequence. Regarding the frame encoder $\mathbb{E}$, we decided to keep the visual encoder frozen throughout all training phases, as

fine-tuning the visual encoder even with a small-scale dataset can affect its image representation capabilities and yield performance drop, as discussed in [86].

Followed by [50, 103], the model training is done in two phases. In the first phase, we only focus on fine-tuning the projection layer with a subset of (VIDEO, INSTRUCTION, RESPONSE) pairs that do not include time boundary and object trajectory. During the second phase, both the newly integrated Adapters and the projection layer are trained with the entire dataset with 8 NVIDIA A100 GPUs. our model establishes a strong baseline and lays the groundwork for future research into more accurate temporal-spatial grounding abilities and context modeling.

## 5 EXPERIMENTS

### 5.1 Evaluation Metrics.

Following the evaluation methods [53, 106] for recent LLMs, we use GPT-4 to assess the quality of responses generated by models. Due to the time-consuming nature of evaluating all 7,700 samples across nine models with GPT-4, we adopt a square root sampling strategy, selecting approximately ($\sqrt{7700} \approx 88$) 100 samples as a representative subset. To deepen our analysis, we further sampled 200 additional responses to evaluate the top four performing models and donate result as EAGLE-pool$_2$ Shikra$_2$ BLIP-2$_2$ and EAGLE$_2$ as shown in Table 5. The results from this extended dataset are presented in the subsequent table and are consistent with the findings from our initial sample of 100 responses.

Given the nature of the egocentric dataset, which offers only action labels, recipe steps, and corresponding timestamps, we need to develop ground truth sentences for evaluation purposes. Our empirical findings indicate that compared to using polished sentences of ground truth labels, template-based construction reduces the occurrence of hallucination errors. The evaluation prompt was refined iteratively through trial and error, aiming to improve the accuracy in identifying event boundaries and objects, and to enhance clarity. The evaluation prompt will be included in the supplementary materials.

**Table 4: Video sources and the corresponding number of videos and average actions for training and validation sets.**

| Video sources | Training set | | Validation set | | Total |
|---|---|---|---|---|---|
| | # videos | # actions (avg) | # videos | # actions (avg) | |
| EPIC-KITCHENS [16] | 16,570 (57%) | 4.78 | 2,901 (38%) | 3.98 | 19,471 (53%) |
| Ego4D [23] | 9,050 (31%) | 2.30 | 3,669 (47%) | 2.80 | 12,719 (35%) |
| PTA | 3,355 (12%) | 1.55 | 1,167 (15%) | 1.53 | 4,522 (12%) |
| Total | 28,975 | | 7,737 | | 36,712 |

**Table 5: We evaluated existing models and our EAGLE model. The scores reflect the models' performance in key aspects, with EAGLE achieving the highest scores in Accuracy and Helpfulness, and competitive scores in other areas. Higher scores indicate better performance.**

| Model | Accuracy | Helpfulness | Detail | Conciseness | Consistency | Average |
|---|---|---|---|---|---|---|
| Video-LLaMA [102] | 1.00 | 1.00 | 1.60 | 1.85 | 1.43 | 1.38 |
| LaViLa [105] | 1.17 | 1.15 | 1.95 | 4.63 | 2.73 | 2.33 |
| BLIP-1 [44] | 1.56 | 1.48 | 1.85 | 4.50 | 3.75 | 2.63 |
| LLaVA [50] | 2.81 | 2.9 | 4.56 | 4.12 | 3.38 | 3.55 |
| ImageBind-LLM [25] | 2.96 | 2.97 | 5.45 | 4.64 | 3.71 | 3.95 |
| InstructBLIP [15] | 3.81 | 3.68 | 5.29 | 5.46 | 4.81 | 4.61 |
| Shikra [12] | 4.21 | 4.52 | 6.80 | 4.78 | 5.15 | 5.09 |
| Shikra$_2$ | 4.31 | 4.55 | 6.85 | 4.20 | 5.20 | 5.02 |
| BLIP-2 [43] | 4.62 | 4.78 | 6.14 | 5.51 | 5.53 | 5.32 |
| BLIP-2$_2$ | 4.43 | 4.80 | 6.20 | 5.45 | 5.38 | 5.25 |
| **EAGLE-pool** | 7.13 | 7.32 | 6.52 | 6.45 | 6.10 | 6.70 |
| **EAGLE-pool$_2$** | 7.21 | 7.40 | 6.72 | 6.42 | 6.30 | 6.81 |
| **EAGLE** | **7.32** | **7.51** | **6.90** | **6.75** | 6.65 | **7.03** |
| **EAGLE$_2$** | 7.28 | 7.48 | 6.83 | 6.67 | **6.77** | 7.01 |

These selected responses will be scored by GPT-4 based on five key metrics, each rated on a scale from 1 to 10, with higher scores indicating superior performance. The evaluation metrics are

(1) *Accuracy*: This metric involves assessing if the response reflects the video's content, focusing on activity recognition for EPIC-KITCHENS and Ego4d samples, and the match between predicted and ground truth procedure steps for PTA samples.
(2) *Helpfulness*: evaluating how much the response aids in comprehending the video's content and its broader context. It involves assessing whether the model's output provides actionable insights or clarifies complex elements within the video.
(3) *Level of Detail*: This involves assessing the comprehensiveness and specificity with which the video is described. A high score in this area indicates that the model captures essential objects and events of the video.
(4) *Conciseness*: This metric measures the succinctness and clarity of the response, focusing on delivering essential information without superfluous content. Effective conciseness involves distilling complex information into a clear and brief explanation, which is critical for provide esstial information of the video.
(5) *Consistency*: This assesses the uniformity and reliability of the narrative or description provided by the model across multiple instances or parts of the video.

Details of the responses from different models will be included in the supplementary material.

## 5.2 Baseline Models.

For our baseline models, we use both image-based and video-based approaches. Image-based models include:

(1) *BLIP-2* [43] trained a lightweight Q-Former for multimodal representation alignment and vision-to-language generation, capable of following instructions without multimodal instruction tuning.
(2) *BLIP-1* [44], pre-trained with web data, using a captioner and filter for synthetic captions, excelling in zero-shot video language tasks.
(3) *InstructBLIP* [15], built on BLIP-2, reformats 26 public datasets for instruction tuning, updating only the Q-Former during training.
(4) *LaViLa* [105] is a video narration method that pairs a video encoder with a GPT-2 [71] as language decoder and a T-5 [72] to reduce overfitting and enhance natural language data.
(5) *LLaVA* [50] introduces visual instruction tuning, using GPT-generated data and instructions for conversation, detailed description, and complex reasoning.
(6) *ImageBind-LLM* [25] is an open-source MLLM, with its algorithm details pending publication.
(7) *Shikra* [12] encodes regions in natural language as numerical coordinates to specify regions in user queries.
(8) *Video-LLaMA* [102] trains adapters for aligning video and audio modalities with LLMs, sampling only eight frames from arbitrarily long videos.

Among baseline models, LaViLa is specifically trained on egocentric videos (Ego4d, EPIC-KITCHEN) to generate narrations. Despite this targeted training, our research reveals that in zero-shot learning scenarios, MLLM, in zero-shot setting, outperformed LaViLa for handling egocentric data.

Additionally, to ensure a fair comparison, we chose not to fine-tune the vision encoder in our EAGLE model for egocentric vision adaptation. Instead, we focused on refining the model to improve its spatial-temporal video analysis capabilities. Our findings indicate that our dataset significantly contributes to enhancing the performance of current MLLMs in understanding and interpreting video content.

## 5.3 Results and Analysis

. To validate the performance of EAGLE, we compare it with recent MLLMs [12, 50, 102], on the EAGLE-400K dataset. As Table 5 shows, Shikra and BLIP-2 demonstrate remarkable proficiency, scoring highest in most categories, indicating their reliability, helpfulness, and detailed response capability. Although Video-LLaMA is targeted at video analysis, it exhibits the lowest performance when compared to image-based multimodal large language models (MLLMs), with outputs often arbitrary and failing to capture essential visual information from videos. LLaVA and InstructBLIP demonstrate balanced and above-average performances across all metrics, showcasing their versatility in handling diverse tasks.

Interestingly, while LaViLa is specifically trained on egocentric data, its performance is hindered by its relatively weaker language backbone (GPT-2), resulting in it being outperformed by more advanced MLLMs in a zero-shot setting. This highlights the significant impact that a robust language model can have on performance.

Moreover, ImageBind-LLM excels in providing detailed and consistent responses. This suggests that superior language modeling capabilities, coupled with a more generalized visual encoder, can enhance overall performance significantly.

Comparing the two variants of EAGLE, which utilize different methods for processing video content: Using concatenation of frame features preserves the temporal order of each frame, allowing the model to capture more detailed temporal dynamics and intricate interactions within the video content.EAGLE-pool, on the other hand, employs temporal pooling to aggregate features over time. This approach helps reduce the impact of less relevant information and noise but may also gloss over finer temporal details that are crucial for understanding complex dynamics. Despite these trade-offs, EAGLE-pool still benefits from the extensive EAGLE-400k dataset and performs better than spatial grounding models like Shikra, which focuses more on spatial rather than temporal data.

These scores provide valuable insights into each model's strengths and weaknesses, allowing for informed decisions on their optimal application areas based on specific needs and criteria.

**Ablation Study.** Ablation studies were conducted on the EAGLE-400k dataset using varied training data splits to investigate the impact of spatial and temporal information on egocentric video understanding. The ablation included: removing time boundaries (*w/o time*), excluding object trajectories (*w/o obj*), and eliminating both (*only desc*). As shown in Table 6, Performance tends to decrease when either time or object information is excluded, with the

least effective results observed when relying solely on descriptions. Surprisingly, PTA exhibits the most significant decline in performance when detailed information is removed, indicating procedure learning relies more on temporal and object details.

Table 6: Ablation study with the different split of dataset

| Dataset | EPIC-KITCHEN | Ego4D | PTA |
|---|---|---|---|
| w/o time | 5.9 | 6.1 | 5.9 |
| w/o object | 6.2 | 6.2 | 5.8 |
| only desc | 5.5 | 5.8 | 5.5 |
| all | 6.8 | 6.4 | 6.5 |

## 6 CONCLUSION

In this work, we present the EAGLE-400K dataset and the EAGLE model for holistic egocentric video understanding. The EAGLE-400K dataset consists of 40K question-answer pairs from 36K diverse video clips and EAGLE offers a unified framework for diverse visual computational tasks. We also provide an evaluation method for egocentric vision tasks and demonstrate EAGLE's superior performance. The introduction of a new evaluation metric enhances the understanding of video-based MLLMs. We hope our work can pave the way for augmented reality assistants that aid in complex physical tasks with multimodal perception.

The EAGLE system exhibits a remarkable proficiency in interpreting temporal information from egocentric videos. Despite its impressive capabilities, the system's reliance on human annotation for defining time boundaries and the necessity of a teacher model to generate high-quality reasoning pairs are areas that warrant further exploration. Additionally, the system's limited capacity to identify and track infrequently appearing objects in the dataset is a challenge that needs to be addressed.

Moreover, there's an increased potential for misinformation from model hallucination, where MLLMs might generate plausible but entirely fictitious responses. This can be particularly concerning when models are used to provide feedback or guidance as an AR assistant. The risk is magnified by the model's ability to produce highly realistic and convincing outputs, blurring the line between reality and fiction for users.

## 7 ETHICS STATEMENT

We must admit that the data collection process in our study may inherently carry a certain degree of bias. In an attempt to mitigate this, we have implemented several measures in our pipeline. Initially, we sourced visual data from EPIC-KITCHEN, Ego4D, and PTA datasets, which are collected from a diverse range of sources and are extensively utilized in various research fields. However, we must also consider that the data annotation phase could potentially introduce additional bias, given its dependence on the prior annotations of the source datasets and GPT-4V. To counteract this, we manually sampled and scrutinized the data quality during the generation process from GPT-4V. In the event of identifying any potential issues, we immediately halt the process for a more in-depth investigation.

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
