# OpenReview forum: "EAGLE: Egocentric AGgregated Language-video Engine"
_acmmm.org/ACMMM/2024/Conference — MM2024 Poster_

### Official Review · Reviewer_RveH · 2024-05-23

**Rating:** 4
**Confidence:** 3

**Summary:**

The paper introduces a novel model and dataset for understanding egocentric videos from a first-person perspective. They present EAGLE, a Multimodal Large Language Model (MLLM), and the EAGLE-400K dataset,  the first large-scale instruction-tuning dataset tailored for egocentric video analysis. They also offer a unified framework for various egocentric video understanding tasks, such as action recognition, procedure learning, and moment retrieval. The authors also propose a set of evaluation metrics designed to assess MLLMs for egocentric video understanding and demonstrate EAGLE's superior performance over existing models.

**Strengths:**

Novelty: The paper introduces a new model and dataset that addresses the fragmentation in egocentric video analysis tasks. The EAGLE model's approach to capturing both spatial and temporal information is innovative and provides a more holistic interpretation of video content.

Clarity: The paper is well-structured, with a clear abstract, introduction, related work, detailed description of the dataset and model, experiments, and conclusions. The figures and tables are informative and aid in understanding the content.

**Limitations:**

1. Using the LLMs to build a universal framework for egocentric video analysis is interesting, but how to avoid hallucinations in LLMs, especially, when they understand the video.

2. There seem to be some problems with using GPT4 as an evaluation tool. For example, the language model itself is a probabilistic model, and the output of GPT4 is not necessarily the same every time. How to avoid this problem?

3. Is there a better way to evaluation?

4. In addition to the unified framework, does the EAGLE model have other advantages when dealing with traditional first-person video tasks?

**Suitability:**

3

---

### Official Review · Reviewer_ZxQf · 2024-05-24

**Rating:** 4
**Confidence:** 2

**Summary:**

This paper proposes a video-based multimodal large language model (MLLM), a dataset EAGLE-400K, and annotated PTA dataset for holistic egocentric video understanding.

**Strengths:**

Clarity:

The authors make the paper easy to follow, the contribution and reference are clearly stated.

Experiments:

The experiments are thorough, the authors compare their model to multiple baselines, which gives a clear page of abilities on other models.

Datasets:

1.	The proposed dataset EAGLE-400K has a significantly larger video length than previous datasets, posing a challenge to existing models.
2.	The Annotated PTA dataset fill the gap procedural understanding within current egocentric video datasets.

**Limitations:**

typos：

1.	148 ‘grab a spoon’
2.	213 broken sentence
3.	828 miss placed period

Experiments:

1.	The authors should consider a user study for evaluating their setting of activities. It would be better to see if human preference is consistent with the model.
2.	The proposed model EAGLE is finetuned on the dataset, does other MLLMs also finetuned on the dataset or just directly inference?

Statement:

It is better to make a clearer statement of Multimodal LLM, the experiments are only about egocentric video analysis. For a MLLM, it is also required with other inference abilities and implementation on other tasks, if the authors want to make it an egocentric task, it is proper to make another statement.

**Suitability:**

3

---

### Official Review · Reviewer_Ffqd · 2024-05-24

**Rating:** 4
**Confidence:** 3

**Summary:**

This paper focuses on egocentric video analysis, which involves the importance of understanding human activities from the first-person perspective. In addition, this paper also takes the existing inconsistent datasets and models into consideration. Thus, the EAGLE-400K, a large-scale instruction tuning dataset tailored for egocentric video, is proposed, and a strong video-based multimodal large language model (MLLM) is built to effectively capture both spatial and temporal information. In addition, a comprehensive comparison and discussion are conducted on current state-of-the-art MLLMs, which introduces a new evaluation metric.

**Strengths:**

1) The unifying egocentric video is novel and promising.
2) Given the EPIC-KITCHENS and Ego4D lacks procedure knowledge, The PTA datasets can provide detailed insight into specific procedural tasks, which greatly improves the video understanding.
3) The EAGLE-400K dataset covers multiple datasets with different focuses for a comprehensive dataset.
4) The EAGLE model considers both spatial and temporal information based on the Adapters.

**Limitations:**

1) There are some missing references that need to be corrected.
2) The qualitative results are important and should be partly added into the main paper.
3) Figure 3 needs to be clarified. It should separately mark the spatial and temporal information processing, and explain how the projection, visual encoder, and adapter work.
4) The structured formulation of each example, including the annotated attributes and corresponding values, should be clarified. Additionally, more detailed statistics and analysis should be conducted on the proposed EAGLE-400K dataset and existing datasets.
5) The evaluation prompts are not included in the supplementary material. Additionally, whether these prompts are reasonable and efficient for measuring accuracy, helpfulness, level of detail, conciseness, and consistency is not discussed. It should be noted that helpfulness is a subjective task.

**Suitability:**

2

---

### Official Review · Reviewer_3tWA · 2024-06-06

**Rating:** 3
**Confidence:** 3

**Summary:**

In this paper authors propose EAGLE, a video-base multimodal language model for first-person view scene understanding, and EAGLE-400K, a unified dataset for various egocentric video understanding tasks. A panoramic of related works and existing research and datasets on the field is presented. The proposed model is then compared to existing baselines, empirically showing that it obtains better results and improves the SotA by a certain margin.

**Strengths:**

- The paper is well written
- The quantitative results consistently improve over the SotA

**Limitations:**

- Why is the dataset called EAGLE-400k if it contains 36k videos? This is somehow misleading.
- The criteria used to sample the 100 videos from the dataset should be explicited. Are they randomly chosen or was there a more advanced selection?
- It is not specified whether the trial-and-error approach for prompt generation was used for EAGLE only or for the baseline as well. For a fair comparison, this approach to obtain the best prompt should be applied for all models.
- Some comparison with more recent baseline dating back to the current year may strength the experimental findings, although I assume that most are not peer-reviewed and published yet.
- In the paper it is affirmed that "Shikra and BLIP-2 demonstrate remarkable proficiency, scoring highest in most categories", however in Table 5 your proposal EAGLE obtains much higher scores on every metric. Could authors clarify this claim?
- What kind of biases may the data contain? Authors admit there may be some but overlook describing how they may have affect the results.

I lean towards rejection because, although the proposal is sound, the results are not well justified and in some steps incosistent. Also, no reference to whether the model and the dataset will be released to the public is present, which is fundamental for reproducibility and to contribute to the development of the field.

**Suitability:**

3

---

### Meta-Review · Area_Chair_azQM · 2024-07-01

**Recommendation:** Accept (Poster)
**Confidence:** 4

**Metareview:**

The paper received mixed scores. One reviewer changed the evaluation from BR to WA after rebuttal as some concerns have been cleared. All other reviewers confirmed BA after rebuttal. The average score is on the positive side, however there still are concerns.